# Adaptation of a short and universal learning self-efficacy scale for clinical skills in Turkish

**Alper Bayazit**[1]*, **Ipek Gonullu**[1], **Celal Deha Dogan**[2]

**1** Ankara University Faculty of Medicine, Department of Medical Education and Informatics, Ankara University, Ankara, Turkey, **2** Ankara University Faculty of Educational Sciences, Department of Measurement and Evaluation, Ankara University, Ankara, Turkey

* abayazit@ankara.edu.tr

## Abstract

### Background

The performance of a clinical task depends on an individual's skills, knowledge, and beliefs. However, there is no reliable and valid tool for measuring self-efficacy beliefs toward clinical skills in the Turkish language. This research work aims to study the linguistic equivalence, validity, and reliability of a Self-Efficacy Scale for Clinical Skills (L-SES).

### Materials and methods

After reaching the original item pool of the scale, applying both forward and backward translation processes, and collecting responses of 11 experts from health professional sciences and educational sciences, the translation and adoption processes were completed. We randomly divided 651 medical students' responses to a 15-item questionnaire into two datasets and conducted exploratory factor analysis (EFA) and confirmatory factor analysis (CFA) analyses.

### Results

CFA validated the three-factor model, and the model fit indexes were found to have acceptable values. The item factor loads ranged from .34 to .84, and items in the scale explained 47% of the total variance. Cronbach's alpha (.91), Spearman-Brown (.88), and Guttman Split-Half (.88) coefficients obtained within the scope of internal consistency reliability demonstrated that the scale had the desired internal consistency.

### Conclusion

The Turkish version of the short and universal learning self-efficacy scale for clinical skills questionnaire is a valid and reliable scale for measuring medical students' self-efficacy for clinical skills. Adopted questionnaires may have different factor structures when applied to two different cultures. We also discussed this issue as a hidden pattern in our study.

**Data Availability Statement:** Data are available at Harvard Dataverse: https://doi.org/10.7910/DVN/M0NJIQ.

**Funding:** The author(s) received no specific funding for this work.

**Competing interests:** The authors have declared that no competing interests exist.

## Introduction

Self-efficacy is defined as people's beliefs about their capabilities to produce designated levels of performance [1], including cognitive, affective, and motivational processes, which determine how people feel, think, and motivate themselves, as well as the effect thereof on their behaviors and performances. According to Bandura [2], performing a task is directly related to an individual's necessary skills, knowledge, and beliefs that they can successfully perform the task under challenging circumstances. The theory also states that self-efficacy is a belief about one's capability, which does not match one's actual capability in a specific domain. Considering that most students tend to be overconfident about their capabilities, determining self-efficacy can also offer an opportunity for students to increase their self-awareness. Previous studies have shown that positive self-efficacy perceptions increase student success in education [3]; students' academic self-efficacy beliefs ensure more engaging and effective teaching [4]; measuring self-efficacy can help to identify the students who need additional training [5]. Since health profession education requires having clinical skills and knowledge to gain certain objectives, including cognitive, affective, and psychomotor domains [6, 7] and the ability to cope with difficult situations, researchers have also investigated the relationship between self-efficacy and pathology students' clinical skills [8], medicals students' OSCE (Objective Structured Clinical Examination) performances [9], dental students' orthodontic clinical skills [10], and nursing students' confidence [11]. According to Shahsavari, Ghiyasvandian [12], measuring self-efficacy can create an opportunity for intervention by instructors and may influence perceived anxiety and self-efficacy. Roh, Lim [13] also showed that self-efficacy is an important metric to measure the effectiveness of several teaching methods in health education.

Students' self-efficacy in learning skills for practice-based clinical skills and their beliefs that they can solve it when they encounter a problem can affect their learning processes. In addition, students with this knowledge can test and identify deficiencies in themselves and provide appropriate feedback for their learning processes. For this purpose, Kang, Chang [14] published a new short and universal Learning Self-Efficacy Scale (L-SES) for clinical skills for undergraduate medical students. Monitoring the progress of learners in courses that comprise clinical skills is especially important to plan special studies for students with low learning self-efficacy and adaptive learning environments designed based on the students' characteristics.

In the literature, there is no universal, reliable, and valid tool for measuring the learning self-efficacy for clinical skills in the Turkish language. There is a need for evaluating the students' beliefs regarding their capabilities to solve any clinical problems from different domains; thus, researchers and medical instructors can test students' clinical skills, progress, and factors affecting their confidence during clinical courses and investigate the relationship between self-efficacy and individual differences. The purpose of this study is to adapt the "Short and Universal Learning Self-Efficacy Scale for Clinical Skills" questionnaire to the Turkish language/culture and perform validity and reliability analyses. In addition, an adapted, valid, and reliable self-efficacy scale based on cognitive, affective, and psychomotor domains can benefit instructional designers in creating appropriate learning environments, medical instructors in supporting their students via a valid measurement tool for clinical skills, and medical curriculum developers via providing feedback on medical curricula.

## Materials and methods

### Participants

In this study, we used convenience sampling. However, while selecting the participant student group, the criterion of being a medical student undertaking a clinical skills course was taken

into consideration. A total of 749 students from the Ankara University Faculty of Medicine, Grade-1 and Grade-3 students participated in the adaptation study.

The curriculum of Ankara University Faculty of Medicine, which runs a six-year program, comprises three years of preclinical work followed by three years of clinical work (two years of clerkship and one year of internship). Clinical skills acquisition starts in the Clinical Skills Lab during the first year of medical education. The courses include various simulation practices such as hygienic hand washing, wearing sterile gloves, measuring pulse rate, respiratory rate, and blood pressure, airway opening and airway placement, injections, and history taking, which are performed on models, mannequins, and standardized patients. The students start to practice clinical skills on real patients in the third year and continue practicing during clerkships. The skills referred to as "clinical skills" in this study are the skills acquired by the students, especially before the clerkships. The research sample consists of first- and third-year students whose skills were measured with the Objective Structured Clinical Exam (OSCE) exam.

The final version of the scale adapted to the students was given, and necessary information was provided about the aim of the study and the scale adaptation process. The participants were not requested for their names; information related to only their age, gender, nationality, and period was collected. We excluded 59 student forms with missing values in the questionnaire and 39 responses answered with the same score for all items in the 15-item Likert-type questionnaire. Thus, 651 medical students (50.2% female; 49.8% male) participated in this study. The participants were mostly Grade-1 (48.8%) and Grade-3 (50.5%) students who also take the clinical skills course. Most of the participants were Turkish (92.2%), and others were foreign students who know Turkish very well and are enrolled in courses executed in the Turkish language.

Performing exploratory and confirmatory factor analysis on half of the data is a recommended method, even though there are other conceivable variations of the exploratory and confirmatory factor analysis sequence in a new test development process [15, 16]. Therefore, the responses of 651 participants in the study were randomly divided into two groups, and EFA was applied to one group and CFA to the other group. There are different views on sample size calculation in factor analysis. Pallant [17] states that correlation coefficients obtained from large samples are more reliable than those obtained from small samples. Tabachnick and Fidell [18], stated that if high factor loadings are obtained as a result of the analysis, it may be sufficient for a sample size of 150 participants. In addition to these approaches that specify criterion-based sample calculation, there are studies stating that it should be calculated by the ratio of respondents to the number of items. While Pallant [17] and Nunnally [19] emphasize that there should be 10 respondents per item, some authors consider five times the number of items to be sufficient [18]. Mundfrom, Shaw [20] recommended using a greater variables-to-factors ratio in practice, at least seven if possible. The general opinion is that there are 10–15 respondents per item [21]. There are 15 items on the scale, and we calculated 225 participants for CFA and 225 participants for EFA to be sufficient for our study.

A short informative paragraph was added to the data collection tool to inform the students. In this text, the definition of the term "self-efficacy", the purpose of the data collection tool, and the meaning of "this course" and "clinical skill" expressions are given.

## Questionnaire

The scale was developed by Kang, Chang [14] within the framework of Bloom [22] taxonomy based on a 5-point Likert item from "strongly disagree" to "strongly agree". They produced items after expert consensus and applied the pilot validity and reliability study to 235 medical

students attending the clinical skills course. The first draft of the scale comprised 15 questions, and they removed three items with the consensus of experts taking part in the study and the Content Validity Index (CVI) analysis. The CVI values of the items in the scale comprising 12 questions in the last form vary between .88 and 1. Items in the questionnaire did not show a difference according to gender. In this study, we requested to use the total item pool (15-item version) of the questionnaire and got the authors' permission for the adaptation process.

## Translation and cross-cultural adaptation process

A standardized forward–backward methodology was used for the adaptation process. According to Koller, Aaronson [23], forward translations from English into the target language serve as the first step in the process. Then, a second translator gives the English back-translations of this preliminary translation. When all of the experts' comments and queries have been addressed, the translation is put to the test in a pilot study on a group of volunteers who are native speakers of the target language. The translation process is completed, and a final translation is produced once the comments from the pilot group have been addressed. A total of 15 items in the short and universal learning Self-Efficacy Scale for Clinical Skills were adapted to the Turkish language and Turkish culture, translated from English to Turkish and Turkish to English by specialists. A total of 11 experts from health professional sciences and educational sciences examined and edited both versions of the scale and submitted an online form declaring their scores, recommendations, and comments. During the expert evaluation process, we shared both versions of the items with the original language and their equivalents translated into Turkish. We requested experts' opinions on the translations via a 5-point Likert scale. We stated that if experts gave a score of 3 or less for each item, they should write comments and provide suggestions. According to the experts' suggestions, a questionnaire with a 15-item scale was completed.

## Data analyses

In this study, we performed exploratory factor analysis (EFA) and confirmatory factor analysis (CFA) to investigate and validate the factor structure of the adapted version of the questionnaire. We randomly divided 651 students' responses to a 15-item questionnaire into two equal group datasets. The first dataset comprised 326 student responses, and we applied the Kaiser-Meyer-Olkin test for sampling adequacy and Bartlett's test of sphericity for factor analysis adequacy. In this process, we used the R statistical programming [24] language "psych" package [25] for factor analysis. The other group comprised 325 students' responses that were used for CFA. We then tested the multivariate normality distribution of the variables with the R software "MVN" library [26]. After assessing the multivariate normality, we applied Weighted Least Squares (WLS) approach in LISREL 8.7. We also calculated Cronbach's α correlation coefficient for testing the internal consistency reliability and split-half reliability.

## Ethics

The Ankara University Medical School Clinical Research Ethics Committee approved this research in October 2019 with a decision number of I4-146-19. The consent was written and Ethics Committee approved the Consent Form.

## Results

### Exploratory Factor Analysis (EFA)

We divided the dataset into two groups randomly, and 326 students' responses were included in the EFA to explore the factorial structure of the questionnaire. According to Zulkefly and

**Parallel Analysis Scree Plots**

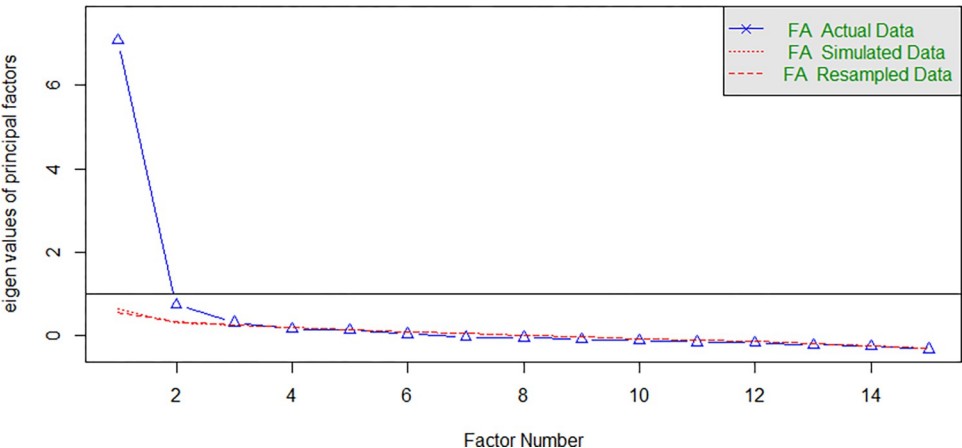

**Fig 1. Parallel analysis scree plots.**

Baharudin [27], Bartlett's test of sphericity is used to test the suitability of the scale for EFA [28]; the Kaiser-Meyer-Olkin (KMO) test result should be higher than .60. The Kaiser-Meyer-Olkin test result of 0.94 verified the sampling adequacy for the analysis. Bartlett's test of sphericity was $\chi^2 (105) = 2777.10$, p $< .001$, showing that the correlation structure is adequate for the factor analyses.

We carried out a parallel analysis with "fa.parallel" from the R-"psych" package to determine the successive eigenvalues. According to Revelle [29], sharp breaks in the plot represent the number of components or factors to extract. The results suggested the number of factors as 3. The eigenvalues of principal factors are given in Fig 1.

We applied the EFA with the "minres" function of the R-"psych" package using MinRes (minimum residual) and "oblimin" rotation, which assumes the factors are correlated. The function aims to minimize the off-diagonal residual correlation matrix by adjusting the eigenvalues of the original correlation matrix. It uses ordinary least squares instead of a maximum likelihood fit function. The exploratory factor analyses' results are presented in Table 1.

As a result of the factor analysis and "Oblimin" rotation, 15 items of the L-SES considered in the analysis come under three factors with eigenvalues greater than 1.0, as seen in Table 1. The variance explained by these three factors regarding the scale is 47%. The first factor had an eigenvalue of 3.284 and an explained variance of 21.9%; the second factor had an eigenvalue of 2.037 and an explained variance of 13.6%; the third factor had an eigenvalue of 1.728 and an explained variance of 11.5%. According to EFA results, the Self-Efficacy Scale for Clinical Skills showed three factors as titled "Cognitive," "Affective" and "Psychomotor." In addition, we also removed Item 11, which showed remarkably similar factor loadings for both Factor 2 ("Affective") and Factor 3 ("Psychomotor").

## Confirmatory Factor Analysis (CFA)

After the removal of Item 11, we applied CFA to the 14-item version of the Self-Efficacy Scale for Clinical Skills with the other half of the dataset. We included 325 students' responses and used the "MVN" package [26] of the R programming language to perform multivariate normality tests. This function performs multivariate skewness and kurtosis tests at the same time and combines test results for multivariate normality. If both tests show multivariate normality, then data follows a multivariate normality distribution at the 0.05 significance level. There was

**Table 1. Exploratory factor analyses results.**

| | Factors and Items | Mean | SD | Factor Loadings | | |
|---|---|---|---|---|---|---|
| | | | | F1 | F2 | F3 |
| | **Factor 1: Cognitive** | | | | | |
| 1 | Klinik beceriyi nasıl yapacağımı hatırlayabilirim. [I can recall how to perform the clinical skill.] | 3.95 | 0.86 | 0.73 | | |
| 2 | Klinik becerinin basamaklarını anlarım ve başkalarına beceriyi yaparak gösterebilirim. [I understand the content of the clinical skill and can demonstrate it to others.] | 3.98 | 0.81 | 0.84 | | |
| 3 | Klinik beceriyi uygulama basamaklarını unuttuğum zaman, akıl yürütme yoluyla bulabilirim. [In case I forget the steps to operate the clinical skill, I can figure things out through reasoning.] | 3.97 | 0.93 | 0.74 | | |
| 4 | Klinik beceriyi ustalıkla uygulayabilirim. [I can masterfully operate the clinical skill.] | 3.48 | 0.92 | 0.74 | | |
| 5 | Klinik becerinin uygulama amacını ve ilkelerini sözel olarak açıklayabilirim. [I can verbally explain the purpose and principle of operating the clinical skill.] | 3.77 | 0.95 | 0.67 | | |
| 6 | Beceride yer alan basamakların sırasını ve diğer basamaklarla olan ilişkisini sözlü olarak açıklayabilirim. [I can verbally explain the sequence and interrelationship between each step.] | 3.69 | 0.88 | 0.54 | 0.31 | |
| | **Factor 2: Affective** | | | | | |
| 7 | Bu derse diğer derslerden daha fazla zaman harcadığımı düşünüyorum. [I think I spend more time on this course than on others.] | 2.92 | 1.14 | | 0.34 | |
| 8 | Bu derste diğer derslere kıyasla performansımda daha fazla ilerleme kaydettiğimi düşünüyorum. [I think I gain more in this course than in others.] | 3.63 | 0.98 | | 0.57 | |
| 9 | Bu dersle ilgili bilgiye daha fazla önem verme eğilimindeyim. [I tend to pay more attention to information related to this course.] | 3.60 | 0.99 | | 0.79 | |
| 10 | Bu dersle ilgili bilgiyi araştırıp bulma eğilimindeyim. [I tend to actively look for information related to this course.] | 3.34 | 1.11 | | 0.72 | |
| | **Factor 3: Psychomotor** | | | | | |
| 11 | Eğiticinin klinik beceriyle ilgili gerçekleştirdiği basamakları ve eylemleri tam olarak tekrarlayabilirim. [I can precisely imitate the instructor's steps and actions of the clinical skill.] | 3.50 | 1.00 | | 0.31 | 0.30 |
| 12 | Klinik becerinin uygulama basamaklarını kolaylıkla baştan sona yapabilirim. [I can smoothly complete the operation steps of the clinical skills.] | 3.52 | 0.93 | | | 0.45 |
| 13 | Klinik becerimi ilerletmek için gelişimimi izlemeye ve değerlendirmeye çalışırım. [I try to monitor my clinical skill for improvements.] | 3.69 | 0.97 | | | 0.69 |
| 14 | Farklı yaklaşımlarla klinik beceriyi uygulamaya çalışırım. [I try to operate the clinical skill through different approaches.] | 3.65 | 0.98 | | | 0.77 |
| 15 | Klinik uygulamalarımı değerlendirmeye çalışırım ve gerektiği zaman uygun düzeltmeler yaparım. [I try to monitor my clinical operations and make proper adjustments as needed.] | 3.80 | 0.95 | | | 0.70 |

no statistically significant multivariate normality distribution. Therefore, we performed the Weighted Least Square (WLS).

According to the EFA results, after the removal of an item (M11), we applied the CFA to the other half of the dataset. The results of the CFA and fit indexes gathered from the 14 items and 325 students' results are as follows: $\chi^2 = 163.75$, (df = 74, p = 0.00), $\chi^2 / df = 2.21$, RMSEA = 0.06, GFI = 0.98, AGFI = 0.98, IFI = 0.98, CFI = 0.98, NFI = 0.97. Fig 2 presents the path diagram. According to these results, $\chi^2 / df = 2.21$ is lower than 3, and RMSEA, which is a measure of absolute fit, showed a "good fit."

## Comparison of high and low group scores

Item analysis based on the difference between the upper 27% and the lower 27% group averages was conducted to determine how sufficient each item in the scale was to distinguish between the individuals in terms of the characteristics they measured. First, we identified the students in the upper 27% (Group 1) and lower 27% (Group 0) groups. Second, we carried out a normality test to compare the groups. The Shapiro-Wilk test showed a significant difference from normality: W (176) = .80, p = .01 for the lower group and W (176) = .95, p = .01 for the higher group. Thus, we carried out the Wilcoxon signed-rank test to compare the item scores of both groups. The test results showed a statistically significant difference (Z = -16.24; p < 0.001). Therefore, the Turkish version of the scale can be used for measuring the self-efficacy beliefs of students toward clinical skills.

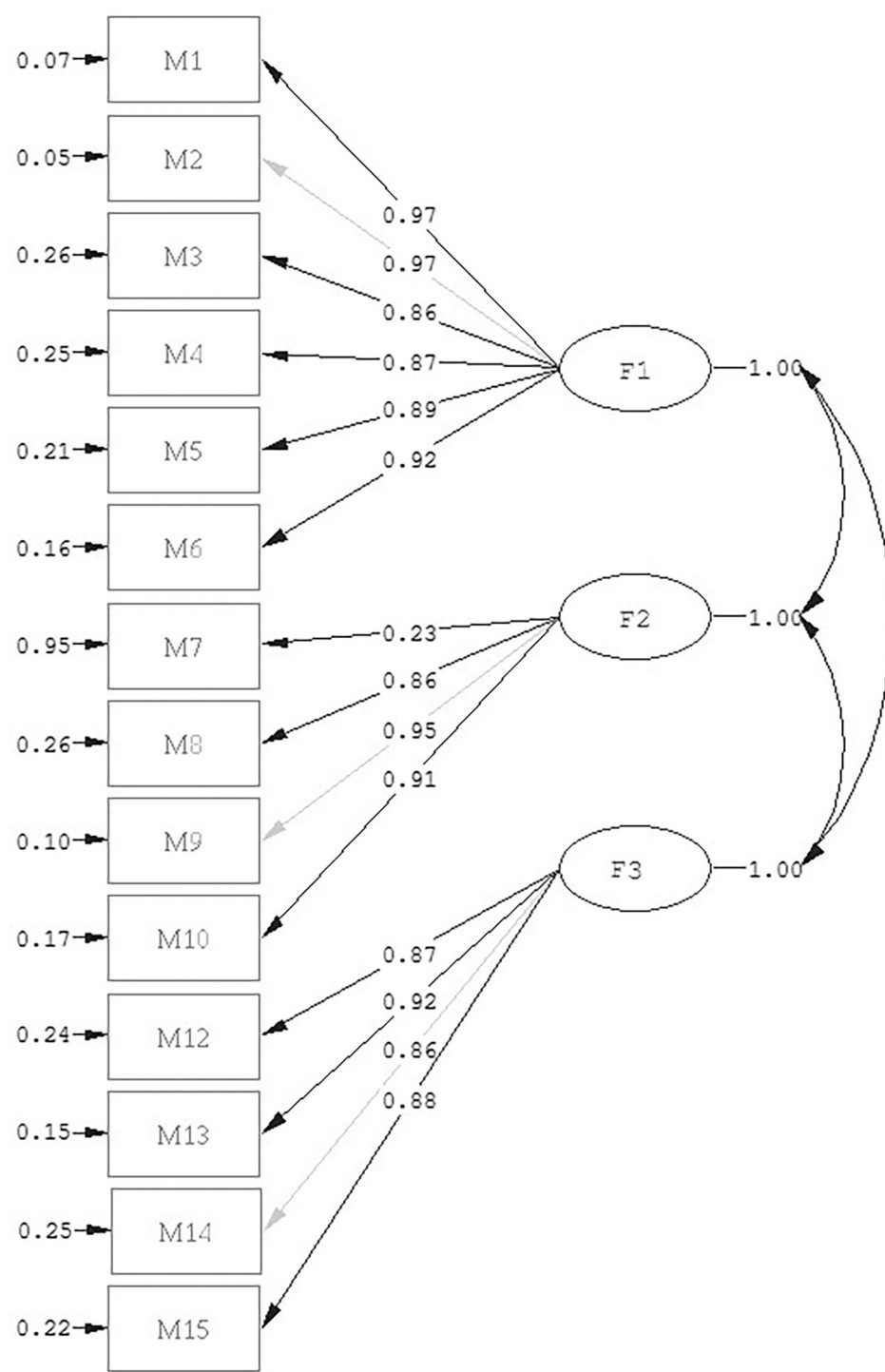

**Fig 2. Confirmatory factor analyses result.**

## Investigation of measurement invariance

Measurement invariance implies that the items of a common test applied in different groups have the same meaning in subgroups [30]. Meredith [31] recommended configural, metric, scalar, and strict invariance as part of multiple group confirmatory factor analysis (MG-CFA). Configural invariance is the first step applied to test the measurement invariance hypotheses, limiting the analysis parameters to the least. The number of factors in the subgroups is expected to be the same, and the indicators of the factor are expected to be similar [32]. Wang and Wang [33] define metric invariance as measuring whether factor loading parameters are invariant in the subgroups to which the test is applied. Thus, different groups interpret similarly the items belonging to the test. The prerequisite for the scalar invariance test is the acceptance of the metric invariance of the relevant test. At this stage, whether the regression constants are also invariant for subgroups is tested [33]. In strict invariance, error variances are also fixed in addition to the presuppositions brought by other sub-invariance types. In this study, we analyzed the measurement invariance in terms of the participants' gender.

Group differences in the means of latent variables can be estimated only if the latent variables are on the same scale in all groups. Thus, the prerequisites for measurement invariance across genders are configural, metric, and scalar across these groups [34]. The configural invariance is satisfied that the basic model structure is invariant across genders (CFI = .94; TLI = .92; SRMR = .05; RMSEA = .09). Similarly, the model fit confirmed the metric invariance across the genders of the participants (CFI = .94; TLI = .93; SRMR = .05; RMSEA = .08; ΔCFI = 0). Furthermore, the scalar invariance was also supported (CFI = .93; TLI = .93; SRMR = .05; RMSEA = .08; ΔCFI = .001). As such, the assumptions of configural, metric, and scalar invariance were satisfied. Strict invariance is not provided (CFI = .93; TLI = .93; SRMR = .06; RMSEA = .08; ΔCFI = .007). A test that provides scalar invariance can be interpreted as containing unbiased items for groups [35]. Therefore, when scalar invariance is provided for a test, comparisons among mean scores can be made between sub-groups. As a result, because invariance is provided, we can say that there is no item biased according to gender in the Turkish version of L-SES, and mean scores can be compared regarding gender.

## Reliability study

**Internal consistency reliability.**   Cronbach's α correlation coefficient was calculated to determine the internal consistency coefficient of the scale. As a result of the analysis, the reliability coefficient was α = 0.91 for the whole scale, α = 0.89 for the "cognitive" dimension, α = 0.69 for the "affective" dimension, and α = 0.85 for the "psychomotor" dimension.

We also examined the consistency of each item with the scale as a whole and tested whether the removal of any question would result in a similar Cronbach's alpha. The removal of Question 11 resulted in a small decrease in Cronbach's alpha, which means that the item is a cause of inconsistency, and we knew the item was under two different factors. Therefore, we deleted Item 11 from the scale.

**Split-half reliability.**   The method splits the form into two halves, and we applied simultaneously the two halves to the participants: correlation (correlation coefficient of the half-test) and reliability estimation between the scores of the subjects from the halves. We investigated first (Items 1–8) and second half (Items 9–15) correlations, as well as odd (1, 3, 5, 7, 9, 11, 13, 15) and even (2, 4, 6, 8, 10, 12, 14) item correlations as two parameters of the split-half reliability. The first–second half correlations produced Cronbach's alpha (.84), Spearman-Brown (.88), and Guttman Split-Half (.88) coefficients, which showed that the scale had the desired internal consistency. Moreover, the odd–even correlations also showed the same results, with

Cronbach's alpha (.84), Spearman-Brown (.88), and Guttman Split-Half (.88) coefficients, supporting the idea that the scale had the desired internal consistency.

## Discussion

In this study, we translated and adopted a short and universal learning self-efficacy scale for clinical skills (L-SES) to the Turkish language. The original scale has a content validity index of between .88 and 1 with 12 items [14]. Its Turkish version also shows high reliability with a coefficient of 0.91 with 14 items. Pooling of the items of the original scale resulted in a different number of items in the adopted version. When designing the questionnaire and conducting studies, paying attention to the item pool is important because of changes in factor structures [36, 37]. The authors confirmed that the first version of the questionnaire included 15 questions, but they removed three questions (Items 3, 4, and 14) that did not meet the threshold. We discussed and reached a consensus with the authors to include these items in our adaptation study.

Another difference between the two scales was the factor loadings of Item 11. The exploratory factor analyses showed that Item 11, which was considered in the "psychomotor domain" in the original scale, was also found related to the "affective domain" in the Turkish culture. We removed this item from our adopted scale because of low factor loading and its relationship with two domains. This study also showed that, in questionnaire adaptation and translation studies, deleted and accepted items in the questionnaires can show different factor loadings in the original and adapted versions. The original study found no difference in the L-SES total score between the male and female participants, and our study, measurement invariance investigation proved that there is no item biased according to gender.

There are several methods for carrying out exploratory factor analysis, such as principal component analyses (PCA), principal axis factoring (PAF), and maximum likelihood factor analyses (MLFA) [38]. While PCA and PAF reduce the dimension—in PCA, components consider the maximum variance in observed variables—PCA investigates the representation of observed correlations between variables by latent factors [39]. However, the MLFA may outperform PAF in cases of unequal loadings within factors and under extraction [40]. If the common variances are small, they may give similar results [38]; depending on the sample size differences, they may give different results. In this study, PAF confirmed the three-factor structure of the adopted version. This result was remarkably similar to the structure of the original scale [14], which also has three factors, including cognitive (Items 1, 2, 3, 4, 5, 6), affective (Items 7, 8, 9, 10), and psychomotor (Items 11, 12, 13, 14, 15) domains. However, we also found another hidden pattern when applying a two-factor structure. We have seen that Turkish culture may also have a two-factor structure and found that Factor 1 (Items 2, 1, 5, 3, 6, 4, 12, 11, 13) was related to "self-efficacy towards clinical skills" and Factor 2 (Items 10, 9, 8, 7) was related to "self-efficacy towards clinical skills course" (Items 15, 14 were ignored because their factor loadings were related to Factors 1 and 2). We also discussed this issue with the authors of the original article. As a result of these discussions, considering that this study aims to achieve adaptation of the original scale, we reported our study with a three-factor structure, which is also similar to the original questionnaire. Nevertheless, this is an interesting issue for proving that adopted questionnaires may have different factor structures when applied to two different cultures [41, 42]. This issue may a limitation of an adaptation study, and we suggest further investigation into the two-factor structure of this questionnaire.

The Turkish version of the short and universal learning self-efficacy scale for clinical skills questionnaire is a valid and reliable scale for measuring medical students' self-efficacy toward clinical skills. Health professionals who undertake clinical skills courses and medical students

who are planning to or already have undertaken a clinical skills course can apply this questionnaire to measure their self-efficacy beliefs toward clinical skills by replacing the quoted phrases with target clinical skills. Further studies can focus especially on clinical skills courses acquired during the COVID-19 pandemic in distance or blended learning environments.

We applied all the scale procedures used in this study to the participants who had prior experience with the Objective Structured Clinical Exam (OSCE) in medical school. Thus, the data collection was performed before the OSCE examination, and all students' responses were collected within their own context, which creates a limitation. Thus, the adapted scale—since we consider it a short and universal scale—should be applied in different times and environments to test students' self-efficacy toward clinical skills. Further, in this study, we used convenient sampling. It would be more appropriate to apply the adapted scale with the participants in fields such as dentistry, physical therapy and rehabilitation, and nursing for the researchers to investigate the generalization of the scale. In addition, since the OSCEs are conducted at the end of the year and the exam schedule of the second-year students was not convenient for the data collection process, they were not included in the sample; only the first- and third-year students participated in this study, which is another limitation of this research.

## Author Contributions

**Conceptualization:** Alper Bayazit, Ipek Gonullu, Celal Deha Dogan.

**Data curation:** Alper Bayazit, Ipek Gonullu, Celal Deha Dogan.

**Formal analysis:** Alper Bayazit, Celal Deha Dogan.

**Investigation:** Alper Bayazit, Ipek Gonullu, Celal Deha Dogan.

**Methodology:** Alper Bayazit, Ipek Gonullu, Celal Deha Dogan.

**Project administration:** Alper Bayazit.

**Resources:** Ipek Gonullu.

**Software:** Celal Deha Dogan.

**Supervision:** Ipek Gonullu, Celal Deha Dogan.

**Validation:** Alper Bayazit, Ipek Gonullu, Celal Deha Dogan.

**Visualization:** Alper Bayazit.

**Writing – original draft:** Alper Bayazit.

**Writing – review & editing:** Ipek Gonullu, Celal Deha Dogan.

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
