## [Decision Letter · Decision Letter 0]

25 Feb 2022

PONE-D-22-01924Adaptation of a Short and Universal Learning Self-Efficacy Scale for Clinical Skills in TurkishPLOS ONE

Dear Dr. Bayazıt,

Thank you for submitting your manuscript to PLOS ONE. After careful consideration, we feel that it has merit but does not fully meet PLOS ONE’s publication criteria as it currently stands. Therefore, we invite you to submit a revised version of the manuscript that addresses the points raised during the review process.

Our reviewers pointed out their suggestions and concerns on your manuscript. Please carefully fulfill these comments, then submit your revised manuscript to the submission system.

We look forward to receiving your revised manuscript.

Kind regards,

Fatih Özden, PhD

Academic Editor

PLOS ONE

Journal Requirements:

3. PLOS ONE has specific requirements for studies that are presenting a new method or tool as the primary focus, including a newly developed or modified questionnaire or scale (https://journals.plos.org/plosone/s/submission-guidelines#loc-methods-software-databases-and-tools.) One requirement is that the questionnaire or scale must be openly available under a license no more restrictive than CC BY. In light of this, before we proceed, please include a copy of your questionnaire or scale as a Supporting Information file (in the original language) or provide a link if it is available through an online repository.

Reviewers' comments:

Reviewer's Responses to Questions

**Comments to the Author**

1. Is the manuscript technically sound, and do the data support the conclusions?

Reviewer #1: Yes

Reviewer #2: No

Reviewer #3: Partly

Reviewer #4: Yes

2. Has the statistical analysis been performed appropriately and rigorously? 

Reviewer #1: Yes

Reviewer #2: Yes

Reviewer #3: I Don't Know

Reviewer #4: Yes

3. Have the authors made all data underlying the findings in their manuscript fully available?

Reviewer #1: Yes

Reviewer #2: Yes

Reviewer #3: Yes

Reviewer #4: No

4. Is the manuscript presented in an intelligible fashion and written in standard English?

Reviewer #1: Yes

Reviewer #2: No

Reviewer #3: Yes

Reviewer #4: Yes

5. Review Comments to the Author

Reviewer #1: Overall, the paper was sound and the statistical methodology seems to me comprehensive. While the bulk of the paper is written in an "intelligible fashion" there are some grammar errors scattered about and a number of sentences that I needed tor read multiple time to try and understand what the authors were trying to state. Therefore, careful editing of the writing to correct errors and enhance clarity is crucial. The authors write: "In this study, we used convenience sampling. However, while selecting the participant student group, the criteria of being a medical student and taking a clinical skills course were taken into consideration." Given the topic of the paper, those criteria would seem to be essential, but I can understand that the authors wanted to state that formally. The authors write: "We also discussed this issue with the authors of the original article. Subsequently…" Is what was done "subsequently" as a result of that discussion? Please be more clear how that discussion influenced the study. In summary, the study itself seems sound enough, methodologically, but it needs to be presented more effectively.

Reviewer #2: It is a simple study; however, the major flaw in this study is that the participants selected do not represent the population needed to be examined, which should be clinical skills.

Grade 1 and Grade 3 students started the clinical skills already? Taking the clinical skills course but not in actual clinical year will not reflect the students' self-efficacy to conduct clinical skills. Thus, the data might be skewed?

What method of forward and backward translation?

Please add sample size calculation for both EFA and CFA.

Why is the limitation in the method section? Please move to the discussion.

Discussion should be based on the previous study and not repeat the results. Please restructure it.

The Figure is very blue, I could not see the numbers.

Reviewer #3: Abstract

In this study, we aimed to adopt linguistic equivalence, validity, and reliability study of a self-efficacy scale for clinical skills (L-SES) developed by Kang, Chang [1]. (No ref in the abstract) they should add in the text (Introduction part and then re number the references accordingly

Introduction

Considering that most students tend to be overconfident about their capabilities [4], ( No need to add as reference , this is a general statement )

The purpose of this study is to adapt the “Short and Universal Learning Self-Efficacy Scale for Clinical Skills” questionnaire into the Turkish language/culture and applying validity and reliability analyses

Materials and Methods

The aim of this research was to carry out a validity and reliability study of the short universal Learning Self-Efficacy Scale developed for clinical skills by adapting it to the Turkish Language and culture. It was mentioned in the Introduction section so need to repeat in this section

The name and surname information were not requested from the participants, only age, gender, nationality, and period information. Why authors need nationality to be included in the demographic section of the questionnaire?

Results

What authors do mean by putting Item 1 Item 2 Item 3 -----------------Item 15

Reviewer #4: This is a well-written and important article. I'm impressed.

A few minor suggestions:

-It is not clear why you have only included Grade 1 and grade 3 students? I think the rationale for choosing the group should also be included. Did the student know about the concept of self-efficacy and if so how they were informed about it? This is important to avoid any kind of research bias.

The limitation section should be after discussion. Please change it.

A complete list of the questions and possible factor analysis would be helpful.

6. PLOS authors have the option to publish the peer review history of their article (what does this mean?). If published, this will include your full peer review and any attached files.

Reviewer #1: No

Reviewer #2: No

Reviewer #3: No

Reviewer #4: No

---

## [Author Response · Author response to Decision Letter 0]

31 Aug 2022

Dear Editor and Reviewers,

On behalf of my co-authors, we thank you very much for giving us an opportunity to revise our manuscript, we appreciate editor and reviewers very much for their constructive comments and suggestions on our manuscript entitled "Adaptation of a Short and Universal Learning Self-Efficacy Scale for Clinical Skills in Turkish". Those comments are all valuable and very helpful for revising and improving our paper. We have studied comments carefully and have made corrections which we hope to meet with approval. Revised portions are marked in yellow in the revised manuscript.

Sincerely,

Authors of the manuscript.

In addition, as a corresponding author, I’m very sorry for very late re-submission, there has been some problems that I’ve not expected. Please accept my apologies for any inconvenience caused.

Yours Sincerely,

Corresponding author.

The following is a point-to-point response to the reviewers’ comments:

Reviewer #1-------------------------------------------------------------

Comment 1: Overall, the paper was sound and the statistical methodology seems to me comprehensive. While the bulk of the paper is written in an "intelligible fashion" there are some grammar errors scattered about and a number of sentences that I needed tor read multiple time to try and understand what the authors were trying to state. Therefore, careful editing of the writing to correct errors and enhance clarity is crucial. The authors write: "In this study, we used convenience sampling. However, while selecting the participant student group, the criteria of being a medical student and taking a clinical skills course were taken into consideration." Given the topic of the paper, those criteria would seem to be essential, but I can understand that the authors wanted to state that formally. The authors write: "We also discussed this issue with the authors of the original article. Subsequently…" Is what was done "subsequently" as a result of that discussion? Please be more clear how that discussion influenced the study. In summary, the study itself seems sound enough, methodologically, but it needs to be presented more effectively.

Answer 1: Thank you very much for the comments and suggestions. We have read the manuscript carefully and re-organized the confusing sentences. In addition, we had another proof reading service and uploaded the Certification file. 

Reviewer #2-------------------------------------------------------------

Comment 1: It is a simple study; however, the major flaw in this study is that the participants selected do not represent the population needed to be examined, which should be clinical skills. Grade 1 and Grade 3 students started the clinical skills already? Taking the clinical skills course but not in actual clinical year will not reflect the students' self-efficacy to conduct clinical skills. Thus, the data might be skewed?

Answer 1: Thank you very much for the comments. Great point. We added an explanation as given below in the Participants section:

The curriculum of Ankara University Faculty of Medicine, which runs a six-year program, comprises three years of preclinical work followed by three years of clinical work (two years of clerkship and one year of internship). Clinical skills acquisition starts in the Clinical Skills Lab during the first year of medical education. The courses include various simulation practices such as hygienic hand washing, wearing sterile gloves, measuring pulse rate, respiratory rate, and blood pressure, airway opening and airway placement, injections, and history taking, which are performed on models, mannequins, and standardized patients. The students start to practice clinical skills on real patients in the third year and continue practicing during clerkships. The skills referred to as “clinical skills” in this study are the skills acquired by the students, especially before the clerkships. The research sample consists of first- and third-year students whose skills were measured with the Objective Structured Clinical Exam (OSCE) exam.

Since the OSCE’ are applied at the end of the year and exam schedule of 2nd year students did not fit to the data collection process, they were not included in the sample. We added this issue as a limitation at the end of the Discussion section.

Comment 2: What method of forward and backward translation?

Answer 2: Thank you for pointing this issue. We used a standardized forward–backward methodology. 

A standardized forward–backward methodology was used for the adaptation process. According to Koller, Aaronson (23), forward translations from English into the target language serve as the first step in the process. Then, a second translator gives the English back-translations of this preliminary translation. When all of the experts’ comments and queries have been addressed, the translation is put to the test in a pilot study on a group of volunteers who are native speakers of the target language. The translation process is completed, and a final translation is produced once the comments from the pilot group have been addressed. A total of 15 items in the short and universal learning Self-Efficacy Scale for Clinical Skills were adapted to the Turkish language and Turkish culture, translated from English to Turkish and Turkish to English by specialists.

The explanation is added to “Translation and Cross-Cultural Adaptation Process” section as suggested.

23. Koller, M., Aaronson, N. K., Blazeby, J., Bottomley, A., Dewolf, L., Fayers, P., ... & EORTC Quality of Life Group. (2007). Translation procedures for standardised quality of life questionnaires: The European Organisation for Research and Treatment of Cancer (EORTC) approach. European Journal of Cancer, 43(12), 1810-1820.

Comment 3: Please add sample size calculation for both EFA and CFA.

Answer 3: Thank you for pointing this issue. 

Performing exploratory and confirmatory factor analysis on half of the data is a recommended method, even though there are other conceivable variations of the exploratory and confirmatory factor analysis sequence in a new test development process [15, 16]. Therefore, the responses of 651 participants in the study were randomly divided into two groups, and EFA was applied to one group and CFA to the other group. There are different views on sample size calculation in factor analysis. Pallant (17) states that correlation coefficients obtained from large samples are more reliable than those obtained from small samples. Tabachnick and Fidell (18), stated that if high factor loadings are obtained as a result of the analysis, it may be sufficient for a sample size of 150 participants. In addition to these approaches that specify criterion-based sample calculation, there are studies stating that it should be calculated by the ratio of respondents to the number of items. While Pallant (17) and Nunnally (19) emphasize that there should be 10 respondents per item, some authors consider five times the number of items to be sufficient [18]. Mundfrom, Shaw (20) recommended using a greater variables-to-factors ratio in practice, at least seven if possible. The general opinion is that there are 10–15 respondents per item [21]. There are 15 items on the scale, and we calculated 225 participants for CFA and 225 participants for EFA to be sufficient for our study.

15. Henson, R. K. ve Roberts, J. K. (2006). Exploratory factor analysis in published research: Common errors and some comment on improved practice. Educational and Psychological Measurement, 66(3), 393-416.

16. Worthington, R. ve Whittaker, T. (2006). Scale development research: A content analysis and recommendations for best practices. Counseling Psychologist, 34, 806-838.

17. Pallant, Julie. (2016). SPSS Survival Manual: a step by step guide to data analysis using IBM SPSS (Ed. 6th). Sydney: George Allen & Unwin

18. Tabachnick, B. G., & Fidell, L. S. (2013). Using multivariate statistics (6th ed.), Boston: Allyn and Bacon

19. Nunnally, J.C. (1978) Psychometric theory. 2nd Edition, McGraw-Hill, New York.

20. Mundfrom, D. J., Shaw, D. G., & Ke, T. L. (2005). Minimum sample size recommendations for conducting factor analyses. International journal of testing, 5(2), 159-168.

21. Pett, M. A., Lackey, N. R., & Sullivan, J. J. (2003). Making sense of factor analysis: The use of factor analysis for instrument development in health care research. sage.

Comment 4: Why is the limitation in the method section? Please move to the discussion.

Answer 4: Thank you for pointing this, we moved it to the discussion section.

Comment 5: Discussion should be based on the previous study and not repeat the results. Please restructure it.

Answer 5: Thank you very much for the suggestion, we restructured the discussion section. We removed the repeated results reporting and added similarities and differences between the original and adopted scales.

Comment 6: The Figure is very blue, I could not see the numbers.

Answer 6: Thank you for pointing this, we updated the figures with higher resolutions.

Reviewer #3-------------------------------------------------------------

Comment 1: In this study, we aimed to adopt linguistic equivalence, validity, and reliability study of a self-efficacy scale for clinical skills (L-SES) developed by Kang, Chang [1]. (No ref in the abstract) they should add in the text (Introduction part and then re number the references accordingly

Answer: Thank you very much for pointing this out. We removed the reference form the abstract, the numbers are updated via EndNote.

Comment 2: Introduction. Considering that most students tend to be overconfident about their capabilities [4], ( No need to add as reference , this is a general statement )

Answer 2: Thank you for pointing this out. We removed the reference.

Comment 3: Materials and Methods. The aim of this research was to carry out a validity and reliability study of the short universal Learning Self-Efficacy Scale developed for clinical skills by adapting it to the Turkish Language and culture. It was mentioned in the Introduction section so need to repeat in this section

Answer 3: Thank you for pointing this out. We removed the repeated paragraph.

Comment 4: The name and surname information were not requested from the participants, only age, gender, nationality, and period information. Why authors need nationality to be included in the demographic section of the questionnaire? 

Answer 4: The students participating in the research study are the students of Medical Education curriculum given in Turkish language. However, since there are immigrants and students from different nationalities in Turkey, the question of nationality was asked in order that it may be needed in further analyzes (for example, to determine the difference in clinical self-efficacy level differences between the immigrants and other students, etc.).

Comment 5: Results. What authors do mean by putting Item 1 Item 2 Item 3 -----------------Item 15

Answer 5: We updated the Table 1 and added both Turkish and English statements by removing “Item”s

Reviewer #4-------------------------------------------------------------

Comment 1: This is a well-written and important article. I'm impressed.

Answer 1: Thank you very much.

Comment 2: It is not clear why you have only included Grade 1 and grade 3 students? I think the rationale for choosing the group should also be included. Did the student know about the concept of self-efficacy and if so how they were informed about it? This is important to avoid any kind of research bias.

Answer 2: Thank you for pointing this out. 

The students were informed in the data collection tool with a short informative paragraph. In this text, the definition of the term “self-efficacy”, the purpose of the data collection tool, and the meaning of "this course" and "clinical skill" expressions are given. This issue is also added to “Participants” section.

Since the OSCEs are conducted at the end of the year and the exam schedule of the second-year students was not convenient for the data collection process, they were not included in the sample; only the first- and third-year students participated in this study, which is another limitation of this research.

We added this information to the limitation section (at the end of the Discussion section).

Comment 3: The limitation section should be after discussion. Please change it.

Answer 3: Thank you for pointing this out. We changed it.

Comment 4: A complete list of the questions and possible factor analysis would be helpful.

Answer 4: Thank you for the suggestion. We updated the Table 1 and added both Turkish and English statements by removing “Item”s as suggested.

Answers to the Editorial Office;

Comment 1: Please provide additional details regarding participant consent. In the ethics statement in the Methods and online submission information, please ensure that you have specified (1) whether consent was informed and (2) what type you obtained (for instance, written or verbal, and if verbal, how it was documented and witnessed). If your study included minors, state whether you obtained consent from parents or guardians. If the need for consent was waived by the ethics committee, please include this information.

ANSWER 1: We collected the consent in written and the Ethics Committee approved the Consent Form. We added this information under the Methods Section, Ethics subheading. Our study does not include the minors. We added this information in both clean and marked up versions of the manuscript files.

Comment 2: PLOS ONE has specific requirements for studies that are presenting a new method or tool as the primary focus, including a newly developed or modified questionnaire or scale (https://journals.plos.org/plosone/s/submission-guidelines#loc-methods-software-databases-and-tools.) One requirement is that the questionnaire or scale must be openly available under a license no more restrictive than CC BY. In light of this, before we proceed, please include a copy of your questionnaire or scale as a Supporting Information file (in the original language) or provide a link if it is available through an online repository.

ANSWER 2: We uploaded the scale (original - Turkish version) as supporting information file

---

## [Decision Letter · Decision Letter 1]

21 Sep 2022

Adaptation of a Short and Universal Learning Self-Efficacy Scale for Clinical Skills in Turkish

PONE-D-22-01924R1

Dear Dr. Bayazıt,

We’re pleased to inform you that your manuscript has been judged scientifically suitable for publication and will be formally accepted for publication once it meets all outstanding technical requirements.

Kind regards,

Fatih Özden, PhD

Academic Editor

PLOS ONE

Additional Editor Comments (optional):

Reviewers' comments:

Reviewer's Responses to Questions

**Comments to the Author**

1. If the authors have adequately addressed your comments raised in a previous round of review and you feel that this manuscript is now acceptable for publication, you may indicate that here to bypass the “Comments to the Author” section, enter your conflict of interest statement in the “Confidential to Editor” section, and submit your "Accept" recommendation.

Reviewer #1: All comments have been addressed

2. Is the manuscript technically sound, and do the data support the conclusions?

Reviewer #1: Yes

3. Has the statistical analysis been performed appropriately and rigorously? 

Reviewer #1: Yes

4. Have the authors made all data underlying the findings in their manuscript fully available?

Reviewer #1: Yes

5. Is the manuscript presented in an intelligible fashion and written in standard English?

Reviewer #1: Yes

6. Review Comments to the Author

Reviewer #1: From my perspective the reviewer comments have been adequately addressed; therefore, the paper us now acceptable for publication.

7. PLOS authors have the option to publish the peer review history of their article (what does this mean?). If published, this will include your full peer review and any attached files.

Reviewer #1: No
